# DFT Studies of Dimethylaminophenyl-Substituted Phthalocyanine and Its Silver Complexes [note 1]

**DOI:** 10.3390/molecules29061344

**Published:** 2024-03-18

**Authors:** Martin Breza

**Affiliations:** Department of Physical Chemistry, Slovak Technical University, Radlinskeho 9, SK-81237 Bratislava, Slovakia; martin.breza@stuba.sk

**Keywords:** B3LYP hybrid functional, solvent effect, geometry optimization, electron structure, spin density, electron transitions

## Abstract

The dimethylaminophenyl-substituted silver phthalocyanine [dmaphPcAg] can be used as a UV-vis photoinitiator for in situ preparation of a silver/polymer nanocomposite. To verify early steps of the supposed mechanism of radical polymerization, we performed quantum chemical calculations of ^m^[dmaphPcAg]^q^ complexes with charges q = +1 to −2 in the two lowest spin states m, of a free ligand and its dehydrogenated/deprotonated products ^m^[dmaphPcH_n_]^q^, n = 2 to 0, q = 0, −1 or −2, in the lowest spin states m. The calculated electronic structures and electron transitions of all the optimized structures in CHCl_3_ solutions are compared with experimental EPR and UV-vis spectra, respectively. The unstable ^3^[dmaphPcAg]^+^ species deduced only from previous EPR spin trap experiments was identified. In addition to ^2^[dmaphPcAg]^0^, our results suggest the coexistence of both reaction intermediates ^1^[dmaphPcAg]^−^ and ^3^[dmaphPcAg]^−^ in reaction solutions. Silver nanoparticle formation is a weak point of the supposed reaction mechanism from the energetic, stereochemistry, and electronic structure points of view.

## 1. Introduction

Phthalocyanine (C_8_H_4_N_2_)_4_H_2_ (PcH_2_) contains four isoindole units connected by nitrogen bridges. The so-obtained ring system has 18 delocalized π electrons responsible for intense absorption (the Soret band around 400 nm and the Q-band in the red/near-infrared region). Therefore, phthalocyanines are used as dyes and pigments [1]. Metal complexes derived from Pc^2−^ have applications in catalysis, solar cells, and photodynamic therapy [1,2,3,4,5,6,7,8,9,10,11,12,13,14,15,16,17].

Very high absorption in the visible range and high chemical resistance implied the synthesis of complexes of various substituted phthalocyanines with central Al, Mg, and Zn atoms [18,19,20,21,22] to be used as photoinitiators for free-radical (FRP) and/or cationic (CP) polymerizations. Recently, Breloy et al. [23] have synthesized a complex of dimethylaminophenyl-substituted phthalocyanine with silver in an unusual oxidation state +II, ^2^[dmaphPcAg]^0^ (see Figure 1. In the used notation, left superscripts denote spin multiplicities m = 2S +1, where S is the total spin angular momentum. The right superscripts denote charges). The species obtained by photoinduced electron transfer reactions of excited ^m^[dmaphPcAg]^0^* initiated FRP both in laminate as well as in the air. Moreover, a polymer/silver nanocomposite with a homogenous narrow size distribution of spherical silver nanoparticles was formed.

Under exposure to LED@385 nm of ^2^[dmaphPcAg]^0^ in CHCl_3_ solutions a sharp decrease in absorbance at 359 and 723 nm is observed simultaneously with splitting of the Q-band and a rapid increase in a band at 450 nm indicating the formation of silver nanoparticles (see Appendix A) [23]. Well-dispersed spherical Ag nanoparticles in the solution (an average diameter of 9 nm) were confirmed by transmission electron microscopy.

A very weak EPR signal of in situ irradiated ^2^[dmaphPcAg]^0^ in CHCl_3_ using LED@385 nm was detected at g = 2.0021 [23]. It can be assigned to an unstable π-radical cation which is probably transformed into an N-centered radical that was detected as an adduct with N-benzylidene-tert-butylamine N-oxide (PBN) or 2,2-dimethyl-3,4-dihydro-2H-pyrrole 1-oxide (DMPO). The carbon- and nitrogen-centered radicals were detected in similar EPR experiments with PBN and DMPO spin-traps in deoxygenated benzene under argon as well [23].

Because radical species are strongly inhibited by oxygen, FRP does not proceed well in air. Unlike acrylate conversions, epoxy conversions are not affected by atmospheric conditions. Another interesting attribute of the above ^2^[dmaphPcAg]^0^ photoinitiating systems is the electron transfer reactions where the central Ag(II) atom is reduced to homogenously dispersed silver nanoparticles in the polymer matrix.

The following ^2^[dmaphPcAg]^0^ photoinitiation mechanism was proposed [23]:

The photoexcited [dmaphPcAg]* is formed via irradiation in the first step
^2^[dmaphPcAg]^0^ + hυ → ^m^[dmaphPcAg]^0^*(1)The reaction between photoexcited and ground-state species leads to the reduction of Ag(II) to Ag(I) and generates the nitrogen-centered cation radical in the second step
^m^[dmaphPcAg]^0^* + ^2^[dmaphPcAg]^0^ → ^n^[dmaphPcAg]^−^ + ^p^[dmaphPcAg]^+•^(2)In the third step, silver nanoparticles and aromatic carbon-centered radicals are formed.
^n^[dmaphPcAg]^−^ → ^2^Ag^0^ + ^r^[dmaphPc]^−•^(3)Subsequently, a hydrogen-transfer reaction between ^p^[dmaphPcAg]^+•^ and ^2^[dmaphPcAg]^0^ leads to dehydrogenated aromatic-derived aminoalkyl radicals ^s^[dmaphPcAg − H]^q•^ and Brønsted photoacids ^1^H^+^ according to equation (4)
^p^[dmaphPcAg]^+•^ + ^2^[dmaphPcAg]^0^ → ^p^[dmaphPcAg]^0^ + ^s^[dmaphPcAg − H]^q•^ + ^1^H^+^(4)The Brønsted acids subsequently initiate ring-opening reactions in epoxides. ^r^[dmaphPc]^−•^ and ^s^[dmaphPcAg − H]^q•^ initiate the FRP of acrylates. 

Nevertheless, the above FRP mechanism (1)–(4) has several limitations. Whereas ^2^[dmaphPcAg]^0^ and ^m^[dmaphPcAg]^0^* have an odd number of electrons and their doublet ground spin state corresponds to natural radicals, ^n^[dmaphPcAg]^−^, ^p^[dmaphPcAg]^+^, and the monodehydrogenated species ^s^[dmaphPcAg − H]^0^ have an even number of electrons and thus can form biradicals. These correspond to singlet or triplet spin states, but the singlet biradicals are not detectable by EPR measurements. Consequently, quantum-chemical calculations are necessary to describe their spin densities. In [23], only quantum-chemical calculations of neutral ^2^[dmaphPcAg]^0^ in doublet spin state and neutral ^1^[dmaphPcH_2_]^0^ in singlet spin state were performed. In our more recent (TD-)B3LYP study [24], optimal geometries, electronic structure, and electron transitions of ^m^[dmaphPcAg]^q^ species in vacuum with charges q = +1 to −2 in the two lowest spin states m were investigated from the point of view of the Jahn–Teller effect. However, this study was not related to spin distribution in the ^m^[dmaphPcAg]^q^ species.

The aim of our current study is to complete the previous studies on DFT calculations of ^m^[dmaphPcAg]^q^ species in CHCl_3_ with various charges, q, corresponding to silver oxidation states between +III and 0 and in the two lowest spin states, m. We set to describe their electron- and spin-density distributions. To complete the picture, we also studied the dehydrogenated/deprotonated ^1^[dmaphPcH_2_]^0^ species in the lowest spin state. We hope that the obtained results on electronic structure, energetics, and electron transitions will contribute to the verification of the above-mentioned reaction mechanism (1)–(3) [23].

## 2. Results

Geometry optimization of ^m^[dmaphPcAg]^q^ complexes in CHCl_3_ in two lowest spin states without any symmetry restriction (see below) started from their optimized structures in vacuum obtained in [24]. After suitable modifications, these structures were used as starting structures for analogous geometry optimizations of dehydrogenated/deprotonated species ^m^[dmaphPcH_2_]^q^ in the lowest two spin states as well. Although the singlet spin states of the systems under study were treated using an unrestricted formalism (the ‘broken symmetry’ treatment [25]), no spin-polarized solutions were obtained, i.e., their energies are identical to the case of restricted DFT calculations. Gibbs energies and relevant geometry parameters of the stable structures are presented in Table 1, Table 2 and Table 3.

### 2.1. Gibbs Energies 

According to the Gibbs energy data calculated at room temperature (Table 1), the energies of the ^m^[dmaphPcAg]^q^ complexes decrease upon reduction. However, even the structures of the ^m^[dmaphPcAg]^2−^ complexes corresponding to the formal oxidation state Ag(0) seem to be stable. Except for ^3^[dmaphPcAg]^+^, the complexes in lower spin states are more stable. It implies that the ^3^[dmaphPcAg]^+^ biradical can be present in non-vanishing concentrations in the reaction system. Consequently, reaction (2) is correct. Despite the relative concentrations of the deexcitation products of the excited ^m^[dmaphPcAg]^0^* species not necessarily satisfy the Boltzmann distribution law, our results indicate that only ^1^[dmaphPcAg]^−^ and ^3^[dmaphPcAg]^−^ species can co-exist in comparable concentrations. In equilibria, the remaining ^m^[dmaphPcAg]^q^ complexes with the same charges are present only in the more stable form because of the extremely large energy difference between their spin states. Therefore, if the LED@385 nm irradiation (corresponding to 310.7 kJ/mol energy) is fully absorbed by ^2^[dmaphPcAg]^0^ excitation, reaction (2) is shifted to the right in agreement with [23] (the reaction Gibbs energy of −26.5 kJ/mol at 298 K).

According to the data in Table 1, deprotonated ^m^[dmaphPcH_n_]^q^ species, n = 2 or 1, seem to be more stable than their dehydrogenated counterparts (i.e., with q preserved). However, their relative stability is also dependent on the reactions of their formation. The reaction Gibbs energy of ^2^Ag^0^ formation (with atomic Gibbs energy of −146.98864 Hartree) according to reaction (3) is highly positive (+235.0 kJ/mol for ^3^[dmaphPcAg]^−^ and +240.7 kJ/mol for ^1^[dmaphPcAg]^−^) and its equilibrium is shifted right due to subsequent formation and precipitation of silver nanoparticles. Due to the lack of necessary data, we deal only with the (1)–(3) reaction equilibria.

### 2.2. DFT-Optimized Structures 

The DFT-optimized geometries of the ^m^[dmaphPcAg]^q^ complexes are very similar to the ^2^[dmaphPcAg]^0^ structure presented in Figure 1. Except for the dimethylphenyl groups, all ^m^[dmaphPcAg]^q^ structures are planar, only the central Ag atom might be slightly above the plane of four pyridine nitrogen atoms N_py_ (up to 0.019 Å in ^3^[dmaphPcAg]^+^, see Table 2). The values of the lengths of the Ag—N_py_ bonds, as well as of the N_py_—Ag—N_py_ angles indicate that the C_4_ symmetry axis is preserved in all the silver complexes, except ^4^[dmaphPcAg]^0^, ^1^[dmaphPcAg]^−^, and ^3^[dmaphPcAg]^−^. The Ag—N_py_ bond lengths increase with complex reduction up to ^1^[dmaphPcAg]^−^ only. Therefore, the electron density transfer to Ag is not related to its out-of-plane movement.

The DFT-optimized structures of ^m^[dmaphPcH_n_]^q^, n = 2 → 0, are presented in Figure 2, Figure 3, Figure 4, Appendix A, and in Table 3. Except for ^1^[dmaphPcH]^−^, their phthalocyanine cores are planar. Their H—N_py_ bond lengths increase during deprotonation/dehydrogenation of the central ring. Due to H—N_py_ bonds, the C_4_ symmetry axis can be observed only in ^2^[dmaphPc]^−^ and ^1^[dmaphPc]^2−^. 

### 2.3. Electronic Structure Characteristics 

The main features of the ^m^[dmaphPcAg]^q^ complexes are presented in Figure 5, Figure 6, Figure 7, Appendix A, and Table 2. Their Ag—N_py_ bond orders and positive Ag charges decrease upon reduction up to ^m^[dmaphPcAg]^−^. Except for complexes in singlet spin states, the d-electron populations of silver atoms are practically constant. The atomic charges of the pyrrole nitrogen N_py_ are more negative than those of the bridging nitrogen N_br_, and the amine nitrogen N_amin_ has even fewer negative charges. During the reduction of the complex, all N charges become even more negative. Positive charges of carbon atoms at the 4- and 7-positions of the isoindole units denoted as C_α_ decrease with the reduction of the complex. Similarly, small negative charges of carbon atoms at isoindoles 5- and 6-positions denoted as C_β_ increase with the reduction of the complex. Significantly more negative charges of the C_met_ methyl carbons do not depend on the charge q and spin state m of the ^m^[dmaphPcAg]^q^ complexes. Only N_amin_ and C_met_ charges were not affected by the lower symmetry of the complexes studied.

The highest spin density at Ag atoms decreases only slightly with reduction. N_py_ atoms have ca two–three times lower spin density of the same sign, which rises with reduction. In both cases, the spin density increases with spin multiplicity. Except for ^4^[dmaphPcAg]^2−^, the spin density of the opposite sign at N_br_ atoms is about one order lower and of variable signs. The spin density at C_α_ atoms is relevant only in anionic complexes and in higher spin states. C_β_ atoms have even lower spin density. The spin density at N_amin_ and C_met_ atoms is vanishing. Only the N_br_, N_amin_, and C_met_ charges were not affected by the lower symmetry of the complexes under study. Except for ^3^[dmaphPcAg]^+^, only vanishing spin density can be found in dimethylaminophenyl groups.

The main features of the ^m^[dmaphPcH_n_]^q^ species are presented in Appendix A and Table 3. The H–N_py_ bond orders decrease with dehydrogenation/deprotonation. The same trend is exhibited by the most negative N_py_ charges, whereas the N_br_ ones exhibit the reverse trend. The even less negative N_amin_ charges increase with the negative charge of the whole species. The positive C_α_ charges decrease with the total charge of the species and increase with deprotonation/dehydrogenation. Only small changes in the very small C_β_ charges can be observed. Negative C_met_ charges are constant in all ^m^[dmaphPcH_n_]^q^ species under study and are equal to those of ^m^[dmaphPcAg]^q^.

We had only two ^2^[dmaphPcH_n_]^q^ species with non-zero spin. The small spin densities at the N_py_ and N_br_ atoms are of the same sign, unlike the higher ones at C_α_ atoms. The vanishing spin density is at C_β_ and N_amin_ atoms. No spin density was observed at C_met_ atoms.

### 2.4. TD-DFT Calculated Electron Transitions 

In this section we will compare the TD-DFT-calculated electron transitions of ^m^[dmaphPcAg]^q^ complexes in CHCl_3_ (Appendix A, Table 4) with UV-vis spectra of ^2^[dmaphPcAg]^0^ in CHCl_3_ before and during photolysis (Appendix A, Table 4).

The calculated electron transitions of ^2^[dmaphPcAg]^0^ in CHCl_3_ can explain the peaks of its UV-vis spectra before irradiation at 350, 700, and 730 nm, but not at 300 and 660 nm. It implies the existence of several species in the solution. Two intense electron transitions of ^3^[dmaphPcAg]^+^ calculated at 557 nm (oscillator strength f = 0.3) are not observed in the experimental spectra. This excludes its presence in measurable concentrations. The same holds for ^1^[dmaphPcAg]^+^ due to the intense TD-DFT electron transition at 777 nm (f = 0.3) and 776 nm (f = 0.2), and twice at 612 nm (f = 0.3) that are missing in the experimental spectra. The calculated electron transitions of ^4^[dmaphPcAg]^0^ could explain the peaks at 300, 350 700, and 730 nm, but the electron transitions at 573 (f = 0.2), 568 (f = 0.2), 546 (f = 0.3), and 530 nm (f = 0.2) missing in experimental spectra contradict the presence of this species. The presence of ^1^[dmaphPcAg]^−^ in this solution is in agreement with all UV-vis peaks before irradiation. The presence of ^3^[dmaphPcAg]^−^ in this solution explains the peaks at 350 and 730 nm only but cannot be excluded. The absence of ^2^[dmaphPcAg]^2−^ in this solution is indicated by the missing shoulder corresponding to the very intense electron transitions at 639 (f = 0.7), 633 (f = 0.8), and 628 (f = 1.5) nm. Two TD-DFT electron transitions of ^4^[dmaphPcAg]^2−^ at 812 nm (f = 0.3) should be partially visible (the lower wavelength part of the strong peak) in UV-vis spectra below 800 nm, which is in contradiction with zero absorbance at 800 nm. This is the reason for the exclusion of this species.

Only 340 and 740 nm peaks of UV-vis spectra of ^1^[dmaphPcH_2_]^0^ in CHCl_3_ before irradiation can be ascribed to its calculated electron transitions, which implies the presence of several species in the solution. The agreement between the experimental and calculated spectral parameters is worse than in the case of ^2^[dmaphPcAg]^0^. The 680 nm shoulder can be ascribed to very intense TD-DFT electron transitions of ^1^[dmaphPcH]^−^ and ^1^[dmaphPc]^2−^. The intense electron transitions of ^1^[dmaphPcH]^−^ contribute to the 340 nm peak as well. The less intense electron transitions of ^2^[dmaphPc]^−^ contribute to both 340 and 740 nm peaks of the UV-vis spectra, whereas its electron transitions at 778 (f = 0.2) nm and 777 (f = 0.2) nm can be coincide with the strong 740 nm peak. The presence of ^2^[dmaphPcH]^0^ can be excluded due to the missing experimental peak corresponding to the TD-DFT electron transition at 788 (f = 0.4) nm and the lack of contribution to the remaining UV-vis peaks.

The UV-vis spectra of irradiated ^2^[dmaphPcAg]^0^ in CHCl_3_ have peaks at wavelengths very similar to the ones before irradiation, except the shoulder at 450 nm, which has been attributed to the formation of silver nanoparticles [23]. Their intensities decrease with irradiation time, except for the 450 nm shoulder, which exhibits the reverse trend. As above, the electron transitions of ^2^[dmaphPcAg]^0^ explain the peaks at 350, 700, and 720 nm; the electron transitions of ^1^[dmaphPcAg]^−^ contribute to all peaks except the shoulder at 450 nm, and the electron transitions of ^3^[dmaphPcAg]^−^ might explain the peaks at 350 and 720 nm. TD-DFT electron transitions of ^1^[dmaphPcH_2_]^0^ can contribute to the UV-vis peak at 350 nm, whereas the very intense ones at 749 (f = 1.0) and 736 (f = 0.8) nm are probably overlapped by the 720 nm shoulder. The electron transitions of ^1^[dmaphPcH]^−^ contribute to the shoulder at 660 nm, whereas the electron transitions at 325 (f = 0.9), 327 (f = 0.3), and 330 (f = 0.5) nm are overlapped by strong peaks at 300 and 350 nm. The TD-DFT electron transitions of ^2^[dmaphPc]^−^ contribute to strong peaks at 350 and 720 nm, whereas their less intense analogs at 778 (f = 0.2) nm and 777 (f = 0.2) nm are probably superimposed by the shoulder at 720 nm. Finally, the electron transitions of ^1^[dmaphPc]^2−^ contribute to the shoulder at 660 nm. 

We can conclude that our TD-DFT interpretation of the experimental UV-vis spectra indicates that the ^2^[dmaphPcAg]^0^, ^1^[dmaphPcAg]^−^, ^3^[dmaphPcAg]^−^, ^1^[dmaphPcH_2_]^0^, ^1^[dmaphPcH]^−^_,_ ^2^[dmaphPc]^−^, and ^1^[dmaphPc]^2−^ species may be present in the reaction system under study.

## 3. Discussion

Two- or even three-electron reduction of the ^3^[dmaphPc]^+^ species causes only small changes in the electron and spin population at the central Ag atom, and the same holds for the Ag—N_py_ bonds. The added electrons are prevailingly distributed over the dmaphPc^2−^ ligand only, which is typical for ‘non-innocent’ ligands. This reduction does not cause a shift of Ag from the phthalocyanine plane. However, neutral Ag nanoparticles should be formed within the endothermic reaction (3), which is simultaneously connected with ca. 0.8 electron addition to Ag with its move from the ligand plane. The planarity of the phthalocyanine core of ^2^[dmaphPc]^−^ is preserved. These changes should proceed in several reaction steps, including the aggregation of the [dmaphPcAg]^−^ species in the solution.

The results of our ‘broken symmetry’ DFT calculations indicate the nonexistence of singlet biradicals. According to the energy data, the ^3^[dmaphPcAg]^+^ concentration should dominate over their singlet counterparts, but the UV-vis spectra [23] do not confirm their presence in reaction systems. Based on EPR measurements, this can be explained by the very low stability of ^3^[dmaphPcAg]^+^, which was detected using spin traps only [23]. Moreover, only this species has a non-vanishing spin density at N_amin_ atoms (see Figure 5) as proposed in [23]. The calculated electron transitions of the EPR silent ^1^[dmaphPcAg]^+^ species do not agree with UV-vis spectra [23] and so it is not present in reaction solutions.

The presence of ^2^[dmaphPcAg]^0^ in reaction solutions is in agreement with the DFT and TD-DFT calculations as well as with the EPR and UV-vis measurements [23]. Comparison of UV-vis spectra with the TD-DFT electron transition and the DFT energy data indicate the absence of ^4^[dmaphPcAg]^0^ in reaction solutions.

The coexistence of ^1^[dmaphPcAg]^−^ and ^3^[dmaphPcAg]^−^ in reaction solutions is allowed by their energies and by the agreement of the calculated electron transitions with UV-vis spectra [23]. ^1^[dmaphPcAg]^−^ is EPR silent.

Energy data indicate that the concentration of ^2^[dmaphPcAg]^2−^ should dominate over that of ^4^[dmaphPcAg]^2−^ but the calculated electron transitions of both species do not agree with UV-vis spectra [23], implying their absence in reaction solutions.

Based on TD-DFT electron transitions and UV-vis spectra [23], the EPR silent ^1^[dmaphPcH_2_]^0^ species and its deprotonated products ^1^[dmaphPc]^−^ and ^1^[dmaphPc]^2−^ can be present in reaction solutions. The same holds for the dehydrogenated product ^2^[dmaphPc]^−^ but not for ^2^[dmaphPcH]^0^.

However, the calculated electron transitions cannot explain all the spectral peaks in Appendix A, such as the shoulder at 450 nm. In [23], it was ascribed to silver surface plasmon resonance of silver nanoparticles. Simple TD-DFT calculations cannot describe the corresponding electron transition and vibronic interactions must be included for this purpose. Unfortunately, we are not able to conduct such calculations for technical reasons. 

## 4. Methods

The geometries of ^m^[dmaphPcAg]^q^, with charges q = −2 to +1 in the two lowest spin states (defined by spin multiplicities) m, and ^m^[dmaphPcH_n_]^q^, n = 2 to 0, q = 0, 1, or 2, in the lowest spin states m, in CHCl_3_ solutions were optimized using the standard B3LYP [26] hybrid functional with Grimme’s GD3 dispersion correction [27]. The cc-pVDZ-PP pseudopotential and basis set for Ag [28] and cc-pVDZ basis sets for the remaining atoms [29] were used for this purpose in terms of an unrestricted formalism within the ‘broken symmetry’ treatment [25]. The SMD (Solvation Model based on solute electron Density) modification [30] of the integral equation formalism polarizable continuum model was used to account for solvent effects. The optimized structures were tested for the absence of imaginary vibrations using vibrational analysis. The time-dependent DFT method (TD-DFT) [31] for 90–120 states was used to evaluate the excited state energies and the intensities of the corresponding electron transitions. The electronic structure was evaluated in terms of Natural Bond Orbital (NBO) population analysis [32]. The Gaussian16 [33] program package was used to perform all quantum-chemical calculations.

## 5. Conclusions

This study aimed to supplement the quantum-chemical studies related to the ^2^[dmaphPcAg]^0^ photoinitiator of polymerization reactions in [23]. We investigated the initial steps (1)–(3) of FRP proposed in [23] via TD-DFT interpretation of the EPR and UV-vis measurements by verifying the presence of possible reaction intermediates ^m^[dmaphPcAg]^q^, q = 1 → −2, ^1^[dmaphPcH_2_]^0^ and its deprotonation/dehydrogenation products in the irradiated reaction system. 

Our results suggest the presence of crucial unstable ^3^[dmaphPcAg]^+^ species, which was deduced by EPR spin trap experiments [23]; however, its identification was complicated due to the presence of other similar compounds in reaction solutions. 

In addition to ^2^[dmaphPcAg]^0^, our results suggest the coexistence of both ^1^[dmaphPcAg]^−^ (EPR silent) and ^3^[dmaphPcAg]^−^ species in reaction solutions. Therefore, the supposed intermediates of reactions (1)–(3) can be identified. The presence of the EPR silent species ^1^[dmaphPcH_2_]^0^, ^1^[dmaphPc]^−^, and ^1^[dmaphPc]^2−^, as well as of the radical ^2^[dmaphPc]^−^ in reaction solutions cannot be excluded.

However, the formation of silver nanoparticles by reaction (3) may be a weak point of the proposed reaction mechanism from the energetic, stereochemistry, and electronic structure points of view. Unfortunately, we do not have sufficient information to propose any alternative reaction mechanisms. Further experimental and theoretical studies are desirable in this field.

## Figures and Tables

**Figure 1 molecules-29-01344-f001:**
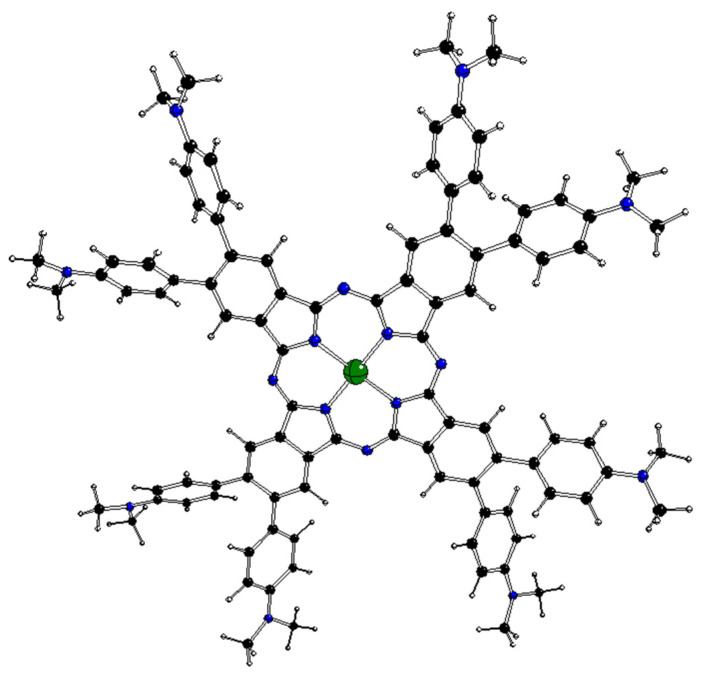
DFT-optimized structure of ^2^[dmaphPcAg]^0^ in CHCl_3_ (C—black, N—blue, Ag—green, H—white).

**Figure 2 molecules-29-01344-f002:**
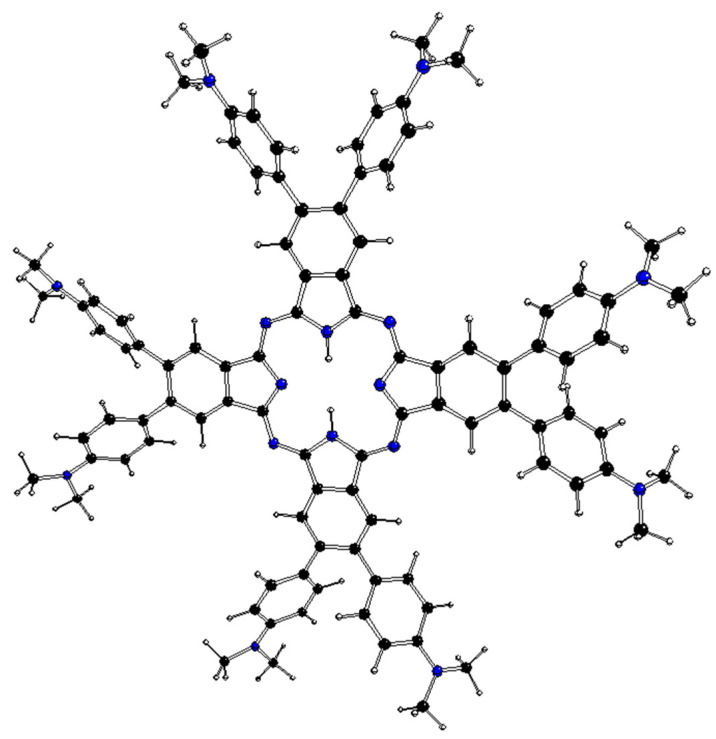
DFT-optimized structure of ^1^[dmaphPcH_2_]^0^ in CHCl_3_ (C—black, N—blue, H—white).

**Figure 3 molecules-29-01344-f003:**
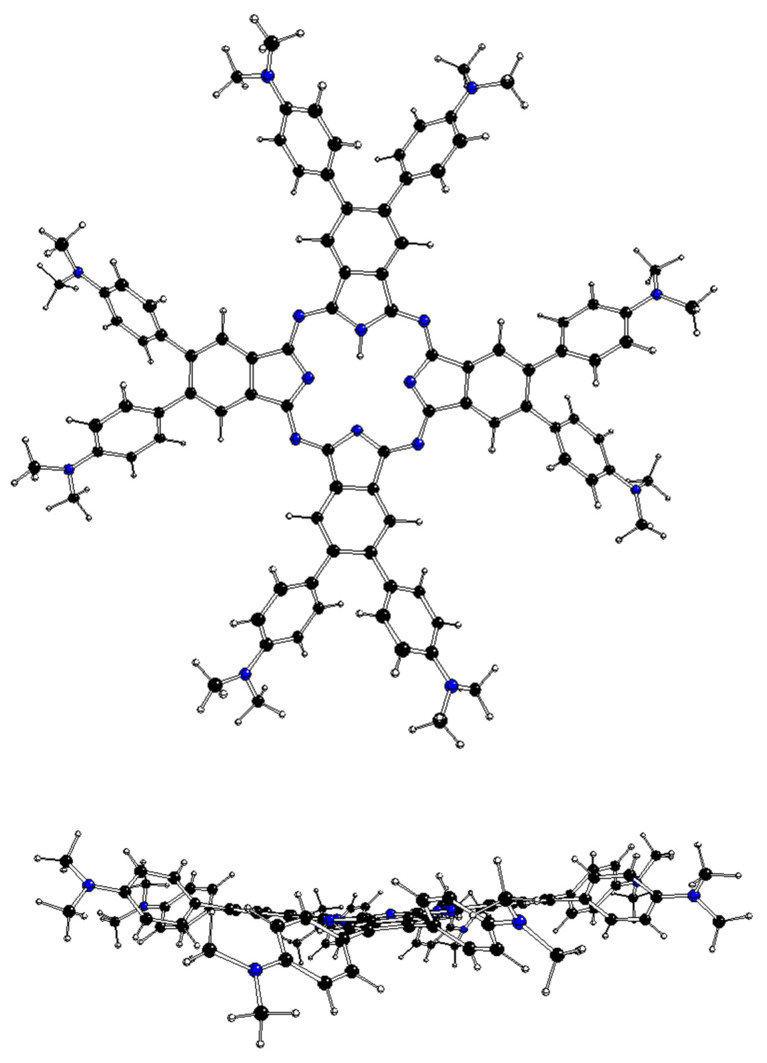
DFT-optimized structure of ^1^[dmaphPcH]^−^ in CHCl_3_—above (**top**) and side (**bottom**) views (C—black, N—blue, H—white).

**Figure 4 molecules-29-01344-f004:**
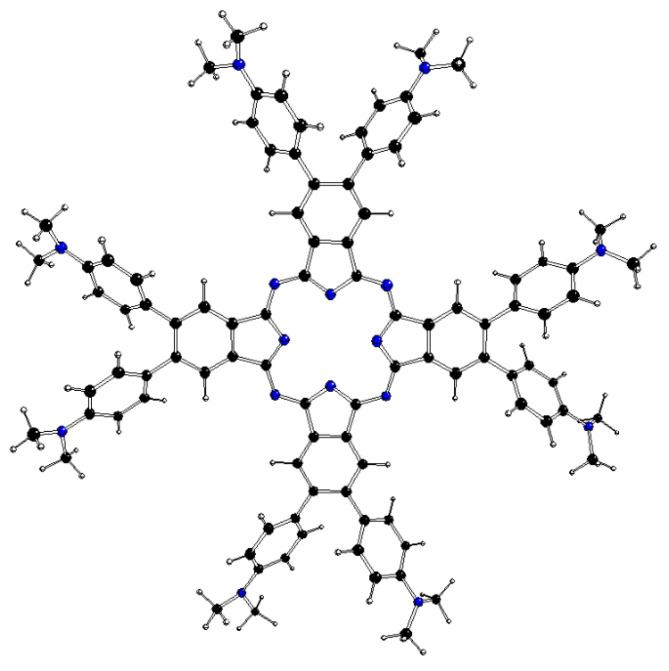
DFT-optimized structure of ^1^[dmaphPc]^2−^ in CHCl_3_ (C—black, N—blue, H—white).

**Figure 5 molecules-29-01344-f005:**
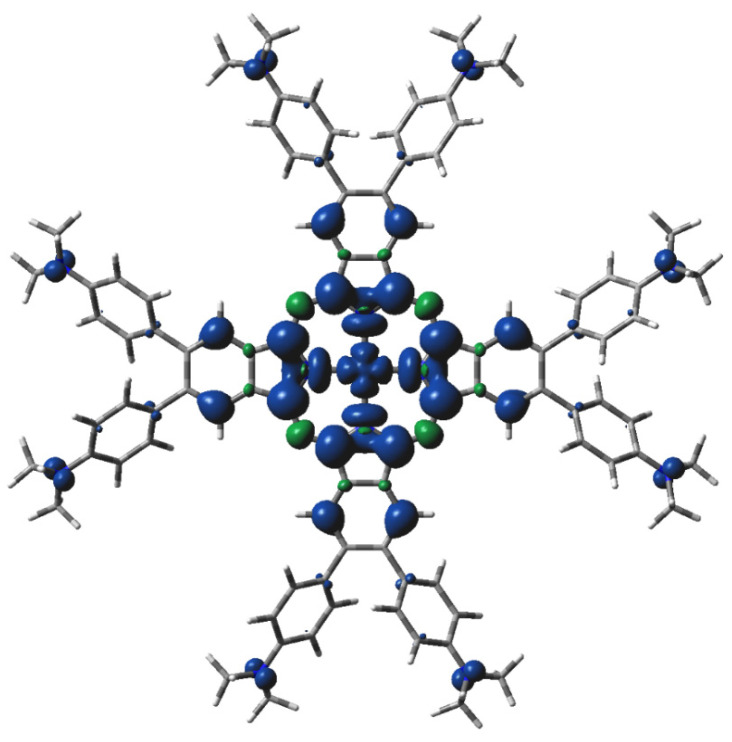
DFT-calculated spin density of ^3^[dmaphPcAg]^+^ in CHCl_3_ (0.001 a.u. isosurface, blue—positive, green—negative spin density).

**Figure 6 molecules-29-01344-f006:**
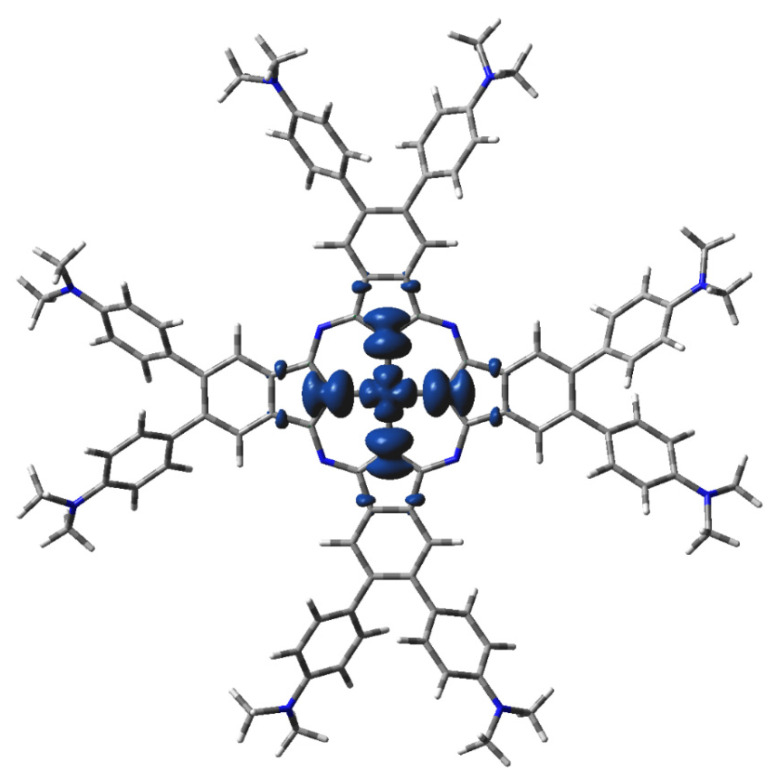
DFT-calculated spin density of ^2^[dmaphPcAg]^0^ in CHCl_3_ (0.001 a.u. isosurface, see Figure 5 for color notation).

**Figure 7 molecules-29-01344-f007:**
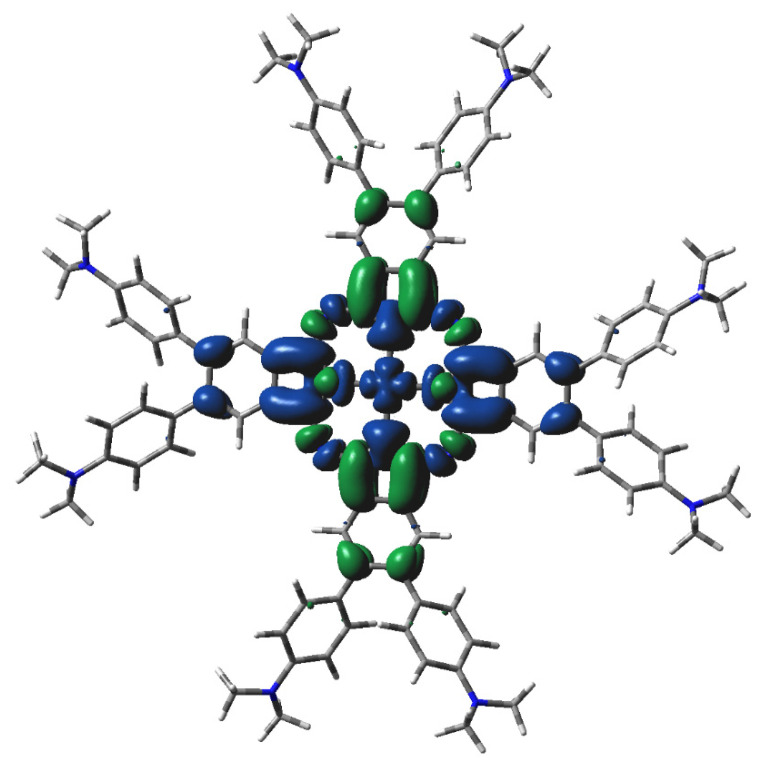
DFT-calculated spin density of ^2^[dmaphPcAg]^2−^ in CHCl_3_ (0.001 a.u. isosurface, see Figure 5 for color notation).

**Table 1 molecules-29-01344-t001:** Absolute (G_298_) and relative (ΔG_298_) Gibbs energies at 298 K of ^m^[dmaphPcAg]^q^ and ^m^[dmaphPcH_n_]^0^ species under study in CHCl_3_; n = 2→0 in various charge (q) and spin (m) states.

Compound	q	m	G_298_ (Hartree)	ΔG_298_ (kJ/mol)
^1^[dmaphPcAg]^+^	+1	1	−4733.24222	453.4
^3^[dmaphPcAg]^+^	+1	3	−4733.25119	429.9
^2^[dmaphPcAg]^0^	0	2	−4733.41492	0.0
^4^[dmaphPcAg]^0^	0	4	−4733.38011	91.4
^1^[dmaphPcAg]^−^	−1	1	−4733.51744	−269.2
^3^[dmaphPcAg]^−^	−1	3	−4733.51527	−263.5
^2^[dmaphPcAg]^2−^	−2	2	−4733.59008	−459.9
^4^[dmaphPcAg]^2−^	−2	4	−4733.57970	−432.6
^1^[dmaphPcH_2_]^0^	0	1	−4587.58495	0.0
^1^[dmaphPcH]^−^	−1	1	−4587.08582	1310.5
^2^[dmaphPcH]^0^	0	2	−4586.94941	1668.6
^1^[dmaphPc]^2−^	−2	1	−4586.54403	2732.9
^2^[dmaphPc]^−^	−1	2	−4586.43712	3013.6

**Table 2 molecules-29-01344-t002:** Relevant geometric and electronic structure parameters of the ^m^[dmaphPcAg]^q^ systems in CHCl_3_ (see Table 1) related to the central Ag, pyrrole N_py_, bridging N_br_, aminyl N_amin_, methyl C_met_, and isoindole C atoms in 4-, 7- (C_α_) and 5-, 6- (C_β_) positions (adj = adjacent, op = opposite bond angles N_py_-Ag-N_py_).

q	+1	+1	0	0	−1	−1	−2	−2
m	1	3	2	4	1	3	2	4
Bond length (Å)								
Ag-N_py_	2.000(4×)	2.056(4×)	2.059(4×)	2.058(2×) 2.062(2×)	2.067(2×) 2.133(2×)	2.063(2×) 2.067(2×)	2.072(4×)	2.071(4×)
Ag—plane distance (Å)								
N_py_ plane	0.001	0.019	0.007	0.000	0.005	0.000	0.000	0.000
Bond angle (deg)								
(N_py_-Ag-N_py_)_adj_	90.0(4×)	90.0(4×)	90.0(4×)	90.0(4×)	89.9(2×) 90.1(2×)	90.0(4×)	90.0(4×)	90.0(4×)
(N_py_-Ag-N_py_)_op_	179.9(2×)	178.9(2×)	179.7(2×)	180.0(2×)	179.2 179.5	180.0(2×)	180.0(2×)	180.0(2×)
Bond order								
Ag-N_py_	0.448(4×)	0.377(4×)	0.379(4×)	0.379(2×) 0.377(2×)	0.363(2×) 0.330(2×)	0.382(2×) 0.381(2×)	0.383(4×)	0.384(4×)
Charge								
Ag	1.023	0.857	0.843	0.833	0.713	0.814	0.787	0.785
N_py_	−0.574(4×)	−0.623(4×)	−0.614(4×)	−0.613(2×) −0.662(2×)	−0.637(2×) −0.614(2×)	−0.603(2×) −0.647(2×)	−0.633(2×) −0.637(2×)	−0.634(4×)
N_br_	−0.533(4×)	−0.553(4×)	−0.550(4×)	−0.587(4×)	−0.566(4×)	−0.579(4×)	−0.617(4×)	−0.610(4×)
C_α_	0.497(8×)	0.528(8×)	0.485(8×)	0.484(4×) 0.532(4×)	0.464(4×) 0.435(4×)	0.422(4×) 0.476(4×)	0.422(8×)	0.418(8×)
C_β_	−0.083(8×)	−0.092(8×)	−0.085(8×)	−0.101(4×) −0.086(4×)	−0.083(4×) −0.091(4×)	−0.100(4×) −0.081(4×)	−0.101(8×)	−0.101(8×)
N_amin_	−0.482(8×)	−0.475(8×)	−0.492(8×)	−0.489(8×)	−0.497(8×)	−0.497(8×)	−0.502(8×)	−0.502(8×)
C_met_	−0.418(16×)	−0.418(16×)	−0.418(16×)	−0.418(16×)	−0.418(16×)	−0.418(16×)	−0.418(16×)	−0.418(16×)
d electron population								
Ag	9.23	9.43	9.43	9.44	9.58	9.44	9.45	9.45
Spin population								
Ag	-	0.426	0.422	0.426	-	0.420	0.412	0.415
N_py_	-	0.119(4×)	0.148(4×)	0.080(2×) 0.206(2×)	-	0.120(2×) 0.234(2×)	0.259(2×) 0.039(2×)	0.208(4×)
N_br_	-	−0.037(4×)	0.002(4×)	0.009(4×)	-	0.054(4×)	−0.001(4×)	0.115(4×)
C_α_	-	0.086(8×)	−0.007(8×)	0.237(4×) 0.100(4×)	-	0.104(4×) −0.016(4×)	−0.106(4×) 0.096(4×)	0.079(8×)
C_β_	-	−0.007(8×)	0.006(8×)	0.016(4×) −0.002(4×)	-	0.037(4×) 0.004(4×)	−0.043(4×) 0.054(4×)	0.047(8×)
N_amin_	-	0.016(8×)	0.000(8×)	0.005(8×)	-	0.000(8×)	0.000(8×)	0.001(4×)
C_met_	-	−0.001(16×)	0.000(16×)	−0.000(16×)	-	0.000(16×)	0.000(16×)	0.000(16×)

**Table 3 molecules-29-01344-t003:** Relevant geometric and electronic structure parameters of ^m^[dmaphPcH_n_]^q^ systems in CHCl_3_ (see Table 1) related to pyrrole N_py_, bridging N_br_, aminyl N_amin_, methyl C_met_, and isoindole C atoms in 4-, 7- (C_α_) and 5-, 6- (C_β_) positions.

Compound	^1^[dmaphPcH_2_]^0^	^1^[dmaphPcH]^−^	^2^[dmaphPcH]^0^	^1^[dmaphPc]^2−^	^2^[dmaphPc]^−^
q	0	−1	0	−2	−1
m	1	1	2	1	2
Bond length (Å)					
H-N_py_	1.017(2×)	1.028	1.029	-	-
Bond order					
H-N_py_	0.659(2×)	0.648	0.645	-	-
Charge					
N_py_	−0.607(2×) −0.656(2×)	−0.577 −0.613 −0.615 −0.611	−0.579 −0.617(2×) −0.628	−0.560(4×)	−0.566(4×)
N_br_	−0.550(4×)	−0.571(2×) −0.565(2×)	−0.566(2×) −0.571(2×)	−0.588(4×)	−0.592(4×)
C_α_	0.488(4×) 0.475(4×)	0.461(2×) 0.462(4×) 0.450(2×)	0.513(2×) 0.519(2×) 0.510(2×) 0.529(2×)	0.434(8×)	0.500(8×)
C_β_	−0.084(4×) −0.090(4×)	−0.087(2×) −0.085(4×) −0.092(2×)	−0.094(2×) −0.089(2×) −0.097(2×) −0.086(2×)	−0.087(8×)	−0.089(8×)
N_amin_	−0.490(4×) −0.492(4×)	−0.497(6×) −0.495(2×)	−0.490(2×) −0.488(4×) −0.486(2×)	−0.503(8×)	−0.496(8×)
C_met_	−0.418(16×)	−0.418(16×)	−0.418(16×)	−0.419(16×)	−0.418(16×)
Spin					
N_py_	-	-	−0.050 −0.042(2×) −0.023	-	−0.049(4×)
N_br_	-	-	−0.050(2×) −0.040(2×)	-	−0.054(4×)
C_α_			0.125(8×)		0.140(8×)
C_β_			−0.007(8×)		−0.002(8×)
N_amin_	-	-	0.004(4×) 0.003(4×)	-	0.002(8×)
C_met_	-	-	0.000(16×)	-	0.000(16×)

**Table 4 molecules-29-01344-t004:** Wavelengths of the measured UV-vis peaks, λ_exp_ (m, medium; s, strong; sh, shoulder) [23], and possible assignments of the TD-DFT-calculated wavelength of the corresponding electron transitions, λ_calc_ (oscillator strengths in parentheses) for the compounds studied in CHCl_3_. The assignments, which are not excluded by missing electron transitions (see text), are denoted in bold.

System	λ_exp_ (nm)	λ_calc_ (nm)	Compound
^2^[dmaphPcAg]^0^	300 m	306(0.3)	^4^[dmaphPcAg]^0^
		**305(0.4)**	**^1^[dmaphPcAg]^−^**
	350 m	**358(0.7), 359(0.8), 346(0.4), 345(0.5)**	**^2^[dmaphPcAg]^0^**
		355(0.7), 350(0.8)	^4^[dmaphPcAg]^0^
		**357(0.3), 355(0.5), 353(0.4), 349(0.5), 342(0.4)**	**^1^[dmaphPcAg]^−^**
		**343(0.8)**	**^3^[dmaphPcAg]^−^**
		**364(1.4), 358(1.0), 349(0.4), 344(0.3), 341(0.5), 333(0.6)**	**^1^[dmaphPcH_2_]^0^**
		**325(0.9), 327(0.3), 330(0.5)**	**^1^[dmaphPcH]^−^**
		**334(0.3)**	**^2^[dmaphPc]^−^**
	660 sh	**668(1.0)**	**^1^[dmaphPcAg]^−^**
		**676(1.2), 672(1.2)**	**^1^[dmaphPcH]^−^**
		**660(1.2), 659(1.2)**	**^1^[dmaphPc]^2−^**
	700 m	701(0.4)	^4^[dmaphPcAg]^0^
		**687(0.7)**	**^1^[dmaphPcAg]^−^**
	730 sh	**724(0.8), 723(0.9)**	**^2^[dmaphPcAg]^0^**
		**724(1.3)**	**^3^[dmaphPcAg]^−^**
		715(0.6)	^4^[dmaphPcAg]^2−^
		**749(1.0), 736(0.8)**	**^1^[dmaphPcH_2_]^0^**
		**722(0.5), 721(0.5)**	**^2^[dmaphPc]^−^**
^1^[dmaphPcH_2_]^0^	340 s	**364(1.4), 358(1.0), 349(0.4), 344(0.3), 341(0.5), 333(0.6)**	**^1^[dmaphPcH_2_]^0^**
		**325(0.9), 327(0.3), 330(0.5)**	**^1^[dmaphPcH]^−^**
		**334(0.3)**	**^2^[dmaphPc]^−^**
	680 sh	**676(1.2), 672(1.2)**	**^1^[dmaphPcH]^−^**
		**660(1.2), 659(1.2)**	**^1^[dmaphPc]^2−^**
	740 s	**749(1.0), 736(0.8)**	**^1^[dmaphPcH_2_]^0^**
		**722(0.5), 721(0.5)**	**^2^[dmaphPc]^−^**
Irradiated reaction system	300 s	**306(0.3)**	**^4^[dmaphPcAg]^0^**
		**305(0.4)**	**^1^[dmaphPcAg]^−^**
	350 s	**358(0.7), 359(0.8), 346(0.4), 345(0.5)**	**^2^[dmaphPcAg]^0^**
		**355(0.7), 350(0.8)**	**^4^[dmaphPcAg]^0^**
		**357(0.3), 355(0.5), 353(0.4), 349(0.5), 342(0.4)**	**^1^[dmaphPcAg]^−^**
		**343(0.8)**	**^3^[dmaphPcAg]^−^**
		**364(1.4), 358(1.0), 349(0.4), 344(0.3), 341(0.5), 333(0.6)**	**^1^[dmaphPcH_2_]^0^**
		**334(0.3)**	**^2^[dmaphPc]^−^**
	450 sh	-	-
	660 sh	**668(1.0)**	**^1^[dmaphPcAg]^−^**
		**676(1.2), 672(1.2)**	**^1^[dmaphPcH]^−^**
		**660(1.2), 659(1.2)**	**^1^[dmaphPc]^2−^**
	700 s	**701(0.4)**	**^4^[dmaphPcAg]^0^**
		**687(0.7)**	**^1^[dmaphPcAg]^−^**
	720 sh	**724(0.8), 723(0.9)**	**^2^[dmaphPcAg]^0^**
		**724(1.3)**	**^3^[dmaphPcAg]^−^**
		715(0.6)	^4^[dmaphPcAg]^2−^
		**722(0.5), 721(0.5)**	**^2^[dmaphPc]^−^**

## Data Availability

Data are contained within the article or Appendix A.

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
