# Peer review of "DFT Studies of Dimethylaminophenyl-Substituted Phthalocyanine and Its Silver Complexes†"

_molecules, 2024, doi:10.3390/molecules29061344_

Round 1

Reviewer 1 Report

Comments and Suggestions for Authors

See the attachment.

Comments on the Quality of English Language

English in this manuscript is good.

Author Response

DFT studies of dimethylaminophenyl-substituted phthalocyanine and of its silver complexes

Martin Breza

Response to Reviewer #1

In this manuscript, the authors utilize dimethylaminophenyl-substituted silver phthalocyanine [dmaphPcAg] as a UV-vis photoinitiator for in situ silver/polymer nanocomposite preparation. Quantum chemical calculations explore the early steps of radical polymerization, revealing identified reaction intermediates and an unstable 3[dmaphPcAg]+ species. EPR and UV-vis spectra comparisons validate calculated electronic structures and electron transitions in CHCl3 solutions. Despite the weak points in silver nanoparticle formation, the study contributes valuable insights into the proposed mechanism's energetic, stereochemical, and electronic aspects. This work needs further improvement before being accepted.

The following suggestions may be considered for reference:

Question 1. The authors added “Figure 1. DFT optimized structure of 2[dmaphPcAg]0 in CHCl3.[23]” in the introduction section. The legend information indicates that it is a DFT-optimized structure, while reference 23 corresponds to an experimentally synthesized structure, causing confusion.

Answer: Ref. 23 contains the results of DFT calculations of 2[dmaphPcAg]0 in CHCl3 as well (geometry optimization, electronic structure, electron transitions, etc.). Therefore, the legend information of Figure 1 is correct.

Question 2. The titles used in the "Results" section of the article are too simple, such as "2.1 Energetics, 2.2 Geometries" and so on.

Answer:  Amended

Question 3. The "Methods" part needs more details. For example, the author shows the DFT optimized structure in Figure 5 and Figure 6. (1) Did the author add restrictive forces to some atoms during the optimization process? (2) The author specifically discusses "2.1 Energetics", then whether the author considers energy correction (dispersion, etc.) in the energy term. If not, please explain the reasons why calibration is not necessary in CHCl3 solvent. (3) (3) Although the SMD solvent model is well discussed in ref. 30, in polar solvents, the PCM model sometimes plays a very critical role, and whether the author explores this work under the PCM solvent model.

Answer: (1) No restrictive forces were applied.

               (2) I do not understand this comment. The use of GD3 dispersion correction with B3LYP calculations is mentioned in 4. Methods.

                (3) According to ref. 30, the SMD model is also parametrized for B3LYP calculations in the CHCl3 solvent. The comparison of various solvent models has been performed in ref. 30 and it is outside the scope of our study.

Question 4. There are too many Figures in the main text. It is recommended that the author combine the Figures organically, such as “Figure *11-16. TD-DFT calculated electron transitions in *** in CHCl3”. Figures 11-16 show TD-DFT calculated electron transitions of different substances in CHCl3. The author used 6 pictures.

Answer:  Figures 11 – 16 are moved to Supplementary.

Reviewer 2 Report

Comments and Suggestions for Authors

For  a non specialist reviewer, this paper presented some problems. For a start, Fig 4 is a direct copy of a Figure in Ref 23 and surely requires permission to reproduce even if the author was one of the authors of the article (same applies to other Figs in the Intro from ref 23) . Moreover it does not help in the comparison with the calculated data of this paper as the positions of the peaks are not given in the Figure. Moreover eq.(1)-(8) are from the same article but have been presented differently, though the original was easier to follow. However, it is hard to see why (4) -(8) are included  as it says on p9 that there is no relevant data to deal with them, so they should be moved to Supp or omitted altogether. It should be noted with regard to eq. (5)-(8) that PhMe is toluene, not what it represents here and in ref..23, and this misused abbreviation should be eliminated.

Are all the data in Tables 2 and 3 needed in the main text . Could some be moved to Supp. Five pages of Introduction is excessive,especially as Figs 1,2,4 are from ref 23 (see copyright permission comment above). The heading for Table 3 appears erroneous as it says the data are for Ag complexes when the entries are for the uncomplexed proligand.

A substantial part of the paper compares the calculated spectra Fig 10-16 with reported data of Figs 2 and 4, but the discussion is hard to follow as Figs 4 and 2 do not have the exact values of peaks marked. Moreover, the discussion switches back and forward between the Fig. 2 and 4, so that a clear picture of how well the calculations match either is hard to follow or the conclusions verified as to which species can be detected. For all their immediate value Figs 10-16 can go to Supp. What is needed is Table comparing calculated data (maxima and intensities) with that of Fig 2 and separately with data derived from Fig 4, so that the basis of the  text conclusions can be understood and verified.  On p15 and elsewhere, absence of bands above 800 nm in the Experimental spectrum 2 is commented on. The experimental spectra cut out at 800 nm, hence such comments are irrelevant. How is the species detected in Fig. 4 which also cuts out at 800 nm. If there is other evidence it should be explained.

In the discussion the last sentence of para 1 does not make sense. Which peaks in Fig. 2 cannot be explained by the calculations as spectra of many species were presented in Figs 10-16. What other species could possibly be there.? This  comment rather undermines the thrust of the paper, as indeed does the penultimate conclusion paragraph.

Comments on the Quality of English Language

Whilst on the whole he English is fine, there ere sentences which do not make sense and need to be reviewed.

Author Response

DFT studies of dimethylaminophenyl-substituted phthalocyanine and of its silver complexes

Martin Breza

Response to Rewiever #2

Question 1. For  a non specialist reviewer, this paper presented some problems. For a start, Fig 4 is a direct copy of a Figure in Ref 23 and surely requires permission to reproduce even if the author was one of the authors of the article (same applies to other Figs in the Intro from ref 23). Moreover it does not help in the comparison with the calculated data of this paper as the positions of the peaks are not given in the Figure. Moreover eq.(1)-(8) are from the same article but have been presented differently, though the original was easier to follow. However, it is hard to see why (4) -(8) are included  as it says on p9 that there is no relevant data to deal with them, so they should be moved to Supp or omitted altogether. It should be noted with regard to eq. (5)-(8) that PhMe is toluene, not what it represents here and in ref..23, and this misused abbreviation should be eliminated.

Answer: Figures 2 and 4 are moved to Supplementary with the standard acknowledgment formulation. The peak positions are summarized in new Table 4 and compared with TD-DFT calculated electron transitions. The remaining figures were not published in ref. 23.

Eqs. (5) – (8) with related text are moved to supplementary and MePh is replaced by tolyl.

Question 2. Are all the data in Tables 2 and 3 needed in the main text . Could some be moved to Supp. Five pages of Introduction is excessive, especially as Figs 1,2,4 are from ref 23 (see copyright permission comment above). The heading for Table 3 appears erroneous as it says the data are for Ag complexes when the entries are for the uncomplexed proligand.

Answer: I have no idea which data in Tables 2 and 3 could be moved to Supplementary.

Figures 2 and 4 are moved to Supplementary with the standard acknowledgment formulation. Fig. 1 was not published in ref. 23.

The heading of Table 3 is corrected.

Question 3. A substantial part of the paper compares the calculated spectra Fig 10-16 with reported data of Figs 2 and 4, but the discussion is hard to follow as Figs 4 and 2 do not have the exact values of peaks marked. Moreover, the discussion switches back and forward between the Fig. 2 and 4, so that a clear picture of how well the calculations match either is hard to follow or the conclusions verified as to which species can be detected. For all their immediate value Figs 10-16 can go to Supp. What is needed is Table comparing calculated data (maxima and intensities) with that of Fig 2 and separately with data derived from Fig 4, so that the basis of the  text conclusions can be understood and verified.  On p15 and elsewhere, absence of bands above 800 nm in the Experimental spectrum 2 is commented on. The experimental spectra cut out at 800 nm, hence such comments are irrelevant. How is the species detected in Fig. 4 which also cuts out at 800 nm. If there is other evidence it should be explained.

Answer: Figs. 2, 4, 10 –16 are moved to Supplementary. The peak positions of Figs. 2 and 4 are summarized in new Table 4 and compared with TD-DFT calculated electron transitions of Figs. 10 - 16. Their discussion is modified in the sense of the rewiever’s comment.

Question 4. In the discussion the last sentence of para 1 does not make sense. Which peaks in Fig. 2 cannot be explained by the calculations as spectra of many species were presented in Figs 10-16. What other species could possibly be there.? This  comment rather undermines the thrust of the paper, as indeed does the penultimate conclusion paragraph.

Answer: The last sentence of the 1st paragraph of Discussion is deleted.

The unexplained 450 nm peak of the irradiated reaction system (possibly ascribed to Ag nanoparticles) is discussed in more detail at the end of Discussion.

Reviewer 3 Report

Comments and Suggestions for Authors

Comments on the manuscript molecules-2879151-v2 by Martin Breza titled “DFT studies of dimethylaminophenyl-substituted phthalocyanine and of its silver complexes”:

The synthesis and properties of dmaphPc and its silver complex ([dmaphPcAg]) were published in ref. [23]. In ref. [24] Breza published additional quantum-chemical calculations for some reaction intermediates [dmaphPcAg]q with charges q = +1 → −2, studied with the aim to explain the possible role of the Jahn–Teller effect. In the study herein, the author presents again DFT computations on m[dmaphPcAg]q species with emphasis on the electron and spin density distributions. The manuscript needs extensive revision. Some shortcomings are listed below:

·       The presence of bis(4-methylphenyl)iodonium hexafluorophosphate (Iod) is not taken into account;

·       I guess the so-called spin state m is actually the multiplicity of the system;

·       It is not clear why the relative Gibbs energies are presented in Table 1, as they do not provide useful information. Particularly for species with different atomic compositions (bottom of Table 1) it is meaningless to calculate relative values. It makes sense to represent the Gibbs energies of the reactions of one form passing into another;

·       Use in the text the IUPAC recommended term "Gibbs energy" instead of "Gibbs free energy";

·       Figure 1 and Figure 2 can be merged into one figure, the same applies to Figure 3 and Figure 4, and to Figures 5-7;

·       Tables 2 and 3 can be presented as SI;

·       2.4. TDDFT calculated electron transitions I recommend to present simulated UV spectra (not transitions) for all structures in a figure - this will allow the reader to directly compare the predicted UV spectra;

·       4. Methods - computational details not to repeat those already published;

·       The author has pushed the results too far and come to questionable conclusions.

Author Response

Author's Notes

The synthesis and properties of dmaphPc and its silver complex ([dmaphPcAg]) were published in ref. [23]. In ref. [24] Breza published additional quantum-chemical calculations for some reaction intermediates [dmaphPcAg]q with charges q = +1 → −2, studied with the aim to explain the possible role of the Jahn–Teller effect. In the study herein, the author presents again DFT computations on m[dmaphPcAg]q species with emphasis on the electron and spin density distributions. The manuscript needs extensive revision. Some shortcomings are listed below:

Question: The presence of bis(4-methylphenyl)iodonium hexafluorophosphate (Iod) is not taken into account;

Answer: It can be seen from the last paragraph of Introduction that this study is restricted to free radical photoinitiation according to the eqs. (1) – (3), i.e. without Iod. Because insufficiently careful reading of this manuscript causes confusions, all mentions of the alternative cationic photopolymerization mechanism using bis(4-methylphenyl)iodonium hexafluorophosphate (Iod) are deleted.

Question: I guess the so-called spin state m is actually the multiplicity of the system;

Answer: It is generally accepted that the spin states of the studied species are defined by the corresponding spin multiplicities. The comment in this sense is added in the 2nd paragraph of Introduction.

Question: It is not clear why the relative Gibbs energies are presented in Table 1, as they do not provide useful information. Particularly for species with different atomic compositions (bottom of Table 1) it is meaningless to calculate relative values. It makes sense to represent the Gibbs energies of the reactions of one form passing into another;

Answer: The relative Gibbs energies are necessary to evaluate the relative abundance of species with the same composition and different charge and/or spin states. According to the Boltzmann law, these species of the same composition with small Gibbs energy differences can coexist in solutions as discussed in Results. As using too many reference energies can cause confusions, only two reference species are used in Table 1.

Question: Use in the text the IUPAC recommended term "Gibbs energy" instead of "Gibbs free energy";

Answer: Amended.

Question: Figure 1 and Figure 2 can be merged into one figure, the same applies to Figure 3 and Figure 4, and to Figures 5-7;

Answer: Refused. Joining the figures proposed cannot improve readability of the manuscript. Moreover, Figure 3 consists of two parts and its joining with Figure 4 should cause confusions.

Question: Tables 2 and 3 can be presented as SI;

Answer: Refused. Both tables are widely discussed in sections 2.3 and 2.4. Their move to SI cannot improve readability of the manuscript.

Question: 2.4. TDDFT calculated electron transitions – I recommend to present simulated UV spectra (not transitions) for all structures in a figure - this will allow the reader to directly compare the predicted UV spectra;

Answer: Refused. The presentation of simulated UV spectra instead of electron transitions shall cause some exact information can be lost.

Question: 4. Methods - computational details not to repeat those already published;

Answer: According to my opinion, all computational details must be summarized in Methods independently of their previous presentation. The readers do not want to waste their time by searching for these details over the entire article.

Question: The author has pushed the results too far and come to questionable conclusions.

Answer: I do not agree. Can you specify which conclusions are questionable? My DFT results successfully explain the reaction mechanism proposed in [23].

Round 2

Reviewer 1 Report

Comments and Suggestions for Authors

I recommend accepting it in its current form.

Author Response

Reviewer has no additional comments.

Reviewer 2 Report

Comments and Suggestions for Authors

Reject

Comments on the Quality of English Language

OK

Author Response

 The revision of this manuscript has not improved it, and in fact it has gone backwards, highlighting further issues.

Question: The representation of molecules as m[ ]q, is inconsistent. When q=0, sometimes ]o is given, but on other occasions no value is shown, and for zero spin o[ is not used but should be for consistency. It is stated that calculations are done for the two lowest spin states, but in Tables 1 and 2, 4 spin states are listed.

Answer: Spin multiplicities m and charges q are added where appropriate. The spin multiplicity m = 0 is nonsense. The species containing an odd number of electrons are calculated with m = 2 and 4, whereas the species containing an even number of electrons are calculated with m = 1 and 3. Therefore, the same system cannot be calculated with all these four spin states (m = 1, 2, 3 and 4).

Question: In the author response, it is stated that Fig 1 is not in ref 23, but in both the original and in the revised version, it is stated that Fig 1 is from ref [23] If Fig 1 and also Fig 2 are both from ref [23], are they needed in the main text?

Answer: Figures 1 and 2 were not published in [23], the citation in their captions is removed.

Question: The main problem arises from attempting to compare the experimental spectra from ref [23] with the data calculated in the current study, as neither experimental nor calculated spectra have exact maxima marked (nor exact intensities). Thus the suggestion was made by a referee to tabulate the exact data and make an appropriate comparison in the text. Unfortunately Table 4 and the text do not fulfil this requirement. For a start, there are constant references to the Supp. Data Figs S1,S2 and S10-17 which have the defects outlined above and make any comparison difficult rather than to Table 4 data.

Answer: Surplus references to figures are deleted.

Question: (In the text there are also references to Figures 2 and 10, which are an uncorrected residue from the initial version.)

Answer: The references are removed.

Question: For the irradiated compound are the data in Table 4 for irradiation in the absence or presence of Iod? Does it matter? This is not clear

Answer:  It is declared in the last paragraph of Introduction that this study is restricted to free radical photoinitiation according to eqs. (1) – (3), i.e. without Iod.  Because insufficiently careful reading of this manuscript can cause confusions, all mentions of the alternative cationic photopolymerization mechanism using Iod are deleted.

Question:In the discussion on pp11,13, it is often not clear whether the non-irradiated or he irradiated solutions are being discussed.

Answer: Rewritten.

Question: In para 2 of the section where the non irradiated system is presumably discussed first, there is reference to an experimental peak at 360 nm, which is not listed in Table 4. In the third para, there is reference to a transition at 557nm which is not in Table 4. Further , calculated features at 777,776, and 612 and later at 573, 568,546, and 530 nm mentioned in the text are not in Table 4. Then in the next paragraph, we have references to an experimental peak at 680 nm, and transitions at 593 and 619 nm not in Table 4. Likewise 639, 633, an 628 of 2[ ]2- are also not in Table 4, and the referenced Fig S17 is not for this species, and data for 4[ ]2-is not found in Fig. S16.

Answer: The purpose of Table 4 is to explain the peaks of the experimental UV-vis spectra by the TD-DFT electron transitions of the individual species. The electron transitions mentioned by the reviewer should correspond to relevant peaks that are not observed in experiment. Therefore, they cannot be presented in Table 4. The table description is specified in this sense.

Question: No reason is given for expecting the 812 nm transition to be observed below 800 nm, i.e within the experimentally observable region.

Answer: Explained in more detail.

Question: The discussion of the transitions for the Ag-free species also contains similar problems where it is not clear whether the discussion refers to the data for the non-radiated or the radiated species.

Answer: Rewritten.

Question: In the last para before the discussion, there is a list of species said to be detected, and whilst such assignments are supported by assignments in Table 4, there are often more than one species assigned to one experimental peak, when the experimental observation could be explained by a single species.

Answer: No single species (except 1[dmaphPcAg]-, which is not realistic) can explain all features of the UV-vis spectra under study.

Question: In the Discussion there is reference to calculations for the 1[ ]+ metallo- species, but no data for this species are in Table 4  

Answer: 1[dmaphPcAg]+ is missing in the solutions studied because its TD-DFT electron transitions do not correspond to any peak in UV-vis spectra.

Reviewer 3 Report

Comments and Suggestions for Authors

Comments on the manuscript molecules-2879151-v4 by Martin Breza titled “DFT studies of dimethylaminophenyl-substituted phthalocyanine and of its silver complexes”:

·       In the previous version of the manuscript "Iod" is mentioned once on page 1, once on page 2 and 5 times on page 3. The author suggests that Iod is involved in reactions with [dmaphPcAg] (lines 55-57: "Fluorescence quenching increases with Iod concentration in a chloroform solution, which suggests an electron transfer reaction from the [dmaphPcAg] excited state to Iod"), so it is reasonable to ask WHY Iod was not considered in the present study, which pretends to deepen the DFT calculations for the m[dmaphPcAg]q species with emphasis on the electron and spin density distributions. Deleting Iod from this version of the manuscript does not solve the problem.

·       My comment about spin and multiplicity is actually taken into account, but the corrections made are not entirely correct: It cannot be said that "The spin states of the studied species are defined by the corresponding spin multiplicities", actually by definition the multiplicity of an energy level is defined as 2S+1, where S is the total spin angular momentum.

Regarding the Author’ answer that ‘’The relative Gibbs energies are necessary to evaluate the relative abundance of species with the same composition and different charge and/or spin states. According to the Boltzmann law, these species of the same composition with small Gibbs energy differences can coexist in solutions as discussed in Results. As using too many reference energies can cause confusions, only two reference species are used in Table 1.

- I cannot accept that values on the order of 3000 kJ/mol are "small energy differences".

·       4. Methods - computational details not to repeat those already published; I note that whole sentences of this section are taken from the previous document (ref.23) (see below).

Polym. Chem., 2021, 12, 1273–1285

molecules-2879151-v4

Standard B3LYP47 geometry optimization with Grimme’s GD3 dispersion correction48 of neutral dmaph-H2 Pc in singlet and dmaph-Ag(II) Pc in doublet ground spin states in chloroform using the cc-pVDZ-PP pseudopotential and basis set for Ag49 and cc-pVDZ basis sets for the remaining atoms50 was performed. Solvent effects were approximated by the SMD modification51 of the integral equation formalism polarizable continuum model. The stability of the optimized structures was confirmed by vibrational analysis (no imaginary vibrations). Excited state energies with the corresponding electron transitions were evaluated using the time-dependent DFT method52–54 for 70 and 99 states, respectively. All calculations were performed using the Gaussian1655 program package.

Standard B3LYP [26] geometry optimization with Grimme’s GD3 dispersion correction [27] of m[dmaphPcAg]q, with charges q = -2 to +1, in two lowest spin states (defined by spin multiplicities) m, and m[dmaphPcHn]q, n = 2 to 0, q = 0, 1, or 2, in the lowest spin states m, in CHCl3 solutions using the cc-pVDZ-PP pseudopotential and basis set for Ag[28] and cc-pVDZ basis sets for the remaining atoms [29] was performed. All calculations were carried out using an unrestricted formalism within ‘broken symmetry’ treatment [25]. Solvent effects were approximated by the SMD (Solvation Model based on solute electron Density) modification [30] of the integral equation formalism polarizable continuum model. The optimized structures were tested on the absence of imaginary vibrations by vibrational analysis. The excited state energies and the intensities of the corresponding electron transitions were evaluated using the time-dependent DFT method (TD-DFT) [31] for 90 - 120 states. The electronic structure was evaluated in terms of Natural Bond Orbital (NBO) population analysis [32]. All calculations were performed using the Gaussian16 [33] program package.

Comments on the Quality of English Language

Some editing of English language required.

Author Response

Comments on the manuscript molecules-2879151-v4 by Martin Breza titled “DFT studies of dimethylaminophenyl-substituted phthalocyanine and of its silver complexes”:

Question:       In the previous version of the manuscript "Iod" is mentioned once on page 1, once on page 2 and 5 times on page 3. The author suggests that Iod is involved in reactions with [dmaphPcAg] (lines 55-57: "Fluorescence quenching increases with Iod concentration in a chloroform solution, which suggests an electron transfer reaction from the [dmaphPcAg] excited state to Iod"), so it is reasonable to ask WHY Iod was not considered in the present study, which pretends to deepen the DFT calculations for the m[dmaphPcAg]q species with emphasis on the electron and spin density distributions. Deleting Iod from this version of the manuscript does not solve the problem.

Answer: In [23] at p. 1277 we have stated that ‘… dmaph-Ag(II)Pc is able to generate photoacids. Notably, the addition of Iod has an incremental effect on the production.’ Thus, the addition of Iod is not necessary for photopolymerization reactions.  As declared in the last sentence of Introduction, the aim of our recent study is to verify the reaction mechanism (1) – (3). Therefore, the DFT study of Iod fluorescence is beyond the scope of this manuscript.

Question:         My comment about spin and multiplicity is actually taken into account, but the corrections made are not entirely correct: It cannot be said that "The spin states of the studied species are defined by the corresponding spin multiplicities", actually by definition the multiplicity of an energy level is defined as 2S+1, where S is the total spin angular momentum.

Answer: The explanation is specified in the sense of this comment.

Question:  Regarding the Author’ answer that ‘’The relative Gibbs energies are necessary to evaluate the relative abundance of species with the same composition and different charge and/or spin states. According to the Boltzmann law, these species of the same composition with small Gibbs energy differences can coexist in solutions as discussed in Results. As using too many reference energies can cause confusions, only two reference species are used in Table 1.

- I cannot accept that values on the order of 3000 kJ/mol are "small energy differences".

Answer: I do not understand this comment. In the 1st paragraph of 2.1 Gibbs energies we state that ‘… only 1[dmaphPcAg]- and 3[dmaphPcAg]- species can co-exist in comparable concentrations. In equilibria, the remaining m[dmaphPcAg]q complexes with the same charges are present only in the more stable form because of the too large energy difference between their spin states.’ The energy difference between the 1[dmaphPcAg]- and 3[dmaphPcAg]- species is only 5.7 kJ/mol (see Table 1).

Question:         4. Methods - computational details not to repeat those already published; I note that whole sentences of this section are taken from the previous document (ref.23) (see below).

Polym. Chem., 2021, 12, 1273–1285

molecules-2879151-v4

Standard B3LYP47 geometry optimization with Grimme’s GD3 dispersion correction48 of neutral dmaph-H2 Pc in singlet and dmaph-Ag(II) Pc in doublet ground spin states in chloroform using the cc-pVDZ-PP pseudopotential and basis set for Ag49 and cc-pVDZ basis sets for the remaining atoms50 was performed. Solvent effects were approximated by the SMD modification51 of the integral equation formalism polarizable continuum model. The stability of the optimized structures was confirmed by vibrational analysis (no imaginary vibrations). Excited state energies with the corresponding electron transitions were evaluated using the time-dependent DFT method52–54 for 70 and 99 states, respectively. All calculations were performed using the Gaussian1655 program package.

Standard B3LYP [26] geometry optimization with Grimme’s GD3 dispersion correction [27] of m[dmaphPcAg]q, with charges q = -2 to +1, in two lowest spin states (defined by spin multiplicities) m, and m[dmaphPcHn]q, n = 2 to 0, q = 0, 1, or 2, in the lowest spin states m, in CHCl3 solutions using the cc-pVDZ-PP pseudopotential and basis set for Ag[28] and cc-pVDZ basis sets for the remaining atoms [29] was performed. All calculations were carried out using an unrestricted formalism within ‘broken symmetry’ treatment [25]. Solvent effects were approximated by the SMD (Solvation Model based on solute electron Density) modification [30] of the integral equation formalism polarizable continuum model. The optimized structures were tested on the absence of imaginary vibrations by vibrational analysis. The excited state energies and the intensities of the corresponding electron transitions were evaluated using the time-dependent DFT method (TD-DFT) [31] for 90 - 120 states. The electronic structure was evaluated in terms of Natural Bond Orbital (NBO) population analysis [32]. All calculations were performed using the Gaussian16 [33] program package.

Answer:   Reformulated.